# Lossless Adaptation of Pretrained Vision Models For Robotic Manipulation

**Mohit Sharma**[*,1,2], **Claudio Fantacci**[2], **Yuxiang Zhou**[2], **Skanda Koppula**[2],
**Nicolas Heess**[2], **Jon Scholz**[2], **Yusuf Aytar**[2]

Carnegie Mellon University[1], DeepMind[2]

## Abstract

Recent works have shown that large models pretrained on common visual learning tasks can provide useful representations for a wide range of specialized perception problems, as well as a variety of robotic manipulation tasks. While prior work on robotic manipulation has predominantly used frozen pretrained features, we demonstrate that in robotics this approach can fail to reach optimal performance, and that fine-tuning of the full model can lead to significantly better results. Unfortunately, fine-tuning disrupts the pretrained visual representation, and causes representational drift towards the fine-tuned task thus leading to a loss of the versatility of the original model. We introduce *lossless adaptation* to address this shortcoming of classical fine-tuning. We demonstrate that appropriate placement of our *parameter efficient adapters* can significantly reduce the performance gap between frozen pretrained representations and full end-to-end fine-tuning without changes to the original representation and thus preserving original capabilities of the pretrained model. We perform a comprehensive investigation across three major model architectures (ViTs, NFNets, and ResNets), supervised (ImageNet-1K classification) and self-supervised pretrained weights (CLIP, BYOL, Visual MAE) in 3 task domains and 35 individual tasks, and demonstrate that our claims are strongly validated in various settings. Please see real world videos at https://sites.google.com/view/robo-adapters.

## 1 Introduction

Pretrained general-purpose vision models, often also referred to as vision foundation models (Yuan et al., 2021), have developed a growing set of perceptual capabilities in recent years. Large-scale vision-language models such as CLIP (Radford et al., 2021) and ALIGN (Jia et al., 2021)) are examples of these highly capable general-purpose vision models which have enabled many applications for image generation/editing (Ramesh et al., 2022; Saharia et al.) and image-based dialog (Alayrac et al., 2022). Existing self-supervised pretrained visual models, such as SimCLR (Chen et al., 2020), BYOL (Grill et al., 2020) or Visual MAE (He et al., 2022), have also been shown to provide strong initializations for a wide range of visual downstream tasks. How can we unlock the power of these models for increasingly novel and challenging control applications?

One solution is to add an output head for each control task and fine-tune the entire architecture. However, fine-tuning degrades performance on the original task(s) the model was trained for, and therefore requires maintaining copies of the model for all tasks we wish to concurrently support. This strategy quickly becomes infeasible as we move towards more general and multi-task agents. For instance, embodied agents acting in the real world will end up solving thousands of downstream manipulation tasks. Given limited hardware capabilities of robots keeping separate copies of increasingly large models (e.g. billions of parameters) for a growing set of tasks is unscalable. This is further exacerbated for robot manipulation wherein hardware and tool differences can result in different task configurations which may require different representations.

In this paper our target is to achieve *lossless adaptation*, which we define as adapting the original pretrained model for the new task or series of tasks, while maintaining the original capabilities of the model. To solve the *lossless adaptation* problem we inject additional parameters, i.e. adapters, to

---

[*]Corresponding author: mohits1@cmu.edu

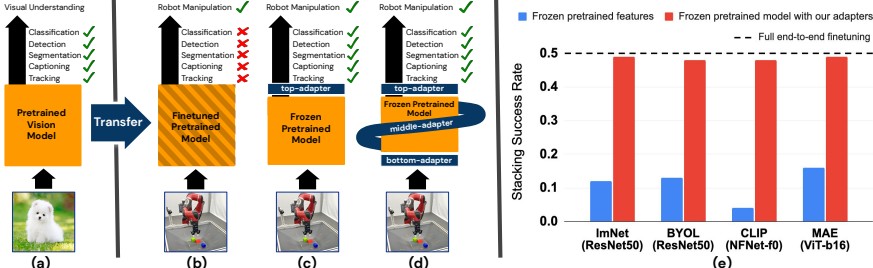

Figure 1: **Parameter efficient lossless adaptation**. Existing works adapt preretrained general purpose visual models (a) through full end-to-end fine-tuning as shown in (b), which looses the original capabilities of the model; or adapting frozen pretrained models through top-adapters as shown in (c), which often fails to achieve optimal control performance. However, by introducing additional mid-level and bottom-level adaptation as in (d), we still maintain the existing perceptual capabilities while approaching the full fine-tuning performance as empirically shown in (e) over many network architectures and pretraining methods.

several specific locations throughout pretrained architecture. We use similar adapters as in previous non-control settings (Rebuffi et al., 2017; Houlsby et al., 2019), but carefully insert them at different network locations to improve off-domain representations for control. We demonstrate that, with a small cost ($\approx 1\%$ of the original model size) of additional parameters scattered throughout the model, we can bridge the performance gap between frozen pretrained features and full end-to-end fine-tuning while maintaining all the original capabilities of a pretrained model (the original model definition can co-exist with the new task head, reusing the vast majority of parameters). We show that as manipulation tasks get harder through complex multi-object interaction and increased level of variation in the randomized initial configurations, the pretrained visual features can't cope with the increased complexity but our parameter efficient adapters can. Overall our contributions include:

- We show that frozen pretrained representations are insufficient to reach optimal manipulation task performance especially for complex manipulation tasks.

- We propose the use of adapters with strong evidence that our adapters can largely close the performance gap between frozen pretrained representations and full end-to-end fine-tuning while adding only a small amount of ($\approx 1\%$ of the original model) adapter parameters.

- Comprehensive evaluation of our approach across 3 different manipulation suites (35 individual tasks), 3 major model architectures (ViTs, NFNets, and ResNets) with supervised (imagenet classification) and self-supervised pretraining (CLIP, BYOL, Visual MAE).

- Experiments demonstrating that adapters also help in sim2real transfer. Thus, enabling fixed large pretrained visual models to be directly used for real world manipulation tasks.

## 2 RELATED WORKS

The general problem of representation learning for control can be broadly divided into two distinct categories – works that use *in-domain* task data to learn task relevant representations for the underlying task, and on the other hand are more recent works that use *out-of-domain* visual data to learn generally useful representations for control.

**In Domain Representation Learning for Control:** Within the first broad category the majority of works learn representations that reflect certain invariances that are presumed to be relevant for the downstream task(s) of interest. Prior works show that useful representations can be learned via data augmentations (Kostrikov et al., 2020; Laskin et al., 2020b), temporal contrastive learning (Laskin et al., 2020a), information bottlenecks (Pacelli & Majumdar, 2020), goal relabeling (Zhou et al., 2021), or via real world priors (Jonschkowski et al., 2017; Jonschkowski & Brock, 2014).

**Out of Domain Representation Learning for Control:** An alternative set of works have emphasized the use of in-the-wild visual datasets for representation learning (Parisi et al., 2022; Shridhar et al., 2022; Khandelwal et al., 2022). They have shown that features learned by large visual models pretrained on common visual learning tasks such as image classification, image inpainting, and contrastive learning can be surprisingly effective for downstream control tasks. Many of these works utilize the large scale pretrained CLIP model (Gadre et al., 2022; Khandelwal et al., 2022; Shridhar

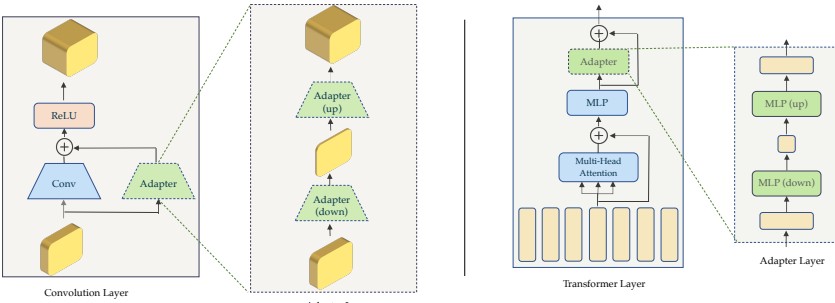

Figure 2: Adapter layers used for convolution based (Left) and transformer based (Right) architectures. For both scenarios we use a bottleneck design.

et al., 2022), while other works also show the effectiveness of features trained on ImageNet (Shah & Kumar, 2021; Parisi et al., 2022), or using temporal data from Ego4D (Nair et al., 2022).

Our work is similar in spirit to above works, i.e., we focus on models trained on large scale out-of-domain data. However, in contrast to prior work which solely shows the effectiveness of the extracted off-the-shelf representations, we aim to adapt those representations for better downstream performance. Concurrent to our work, (Hansen et al., 2022) also show that *fixed* pre-trained representations can be sub-optimal for control. However, where (Hansen et al., 2022) turn to data-augmentation and learning-from-scratch, we demonstrate that adapting pre-trained representations (via full-finetuning or our adpaters) outperforms frozen representations even *without* augmentations. We further establish the advantage of pre-trained representations via sim2real transfer results.

**Transfer Learning:** Adapting a given large model for different downstream tasks has been widely studied in the literature (Rebuffi et al., 2017; Castrejon et al., 2016; Carion et al., 2020; Li et al., 2022; Li & Liang, 2021; Mahabadi et al., 2021; Jia et al., 2022; Sung et al., 2022). Within the vision community Rebuffi et al. (2017) introduced adapter modules to learn a single representation across multiple domains. Li et al. (2022) used similar adapters for few-shot image classification in new domains. However, previous works in the vision community mostly use adapters for classification tasks and the pretraining task is also image classification. By contrast, we focus on control tasks which are significantly different than image classification. In addition to task-differences our work also includes significant domain shifts, e.g. non-canonical camera views, moving robot arm, heavy occlusion, textureless objects. As we show, unlike previous works, this requires carefully inserting adapters in different sections of the network without significantly increasing the parameter count. Adapter modules have also been explored in the language community. Houlsby et al. (2019) adapt the BERT model for different tasks, while Li & Liang (2021) show that optimizing small amount of task-specific vectors (prefixes) can often match the full fine-tuning performance Liu et al. (2021). Another aspect of transfer learning is continual or lifelong learning wherein learning involves a sequence of tasks (Aljundi et al., 2017; Rannen et al., 2017; Rusu et al., 2016; Shin et al., 2017; Schwarz et al., 2018; Yoon et al., 2017), but wherein the original model is distilled for new tasks and thus not necessarily perfectly preserved. Our work is similar to the former set of works – we keep the large pretrained model *frozen* and improve it using light-weight adaptation to extract the final representation for control.

## 3 APPROACH

Our main aim is to use fixed pretrained visual models but adapt their representations for improved downstream control task performance. To achieve this we use parameter efficient adapter modules that can be inserted in appropriate locations throughout the deep network. These non-pretrained adapter modules are the only set of parameters that are updated during downstream policy learning.

A common approach to use fixed pretrained visual models is to attach a learned policy head (i.e. top-adapter) and train for the downstream control task (Nair et al., 2022; Parisi et al., 2022). However such an approach cannot adjust the low-level perception and mid-level abstraction for the down-stream control task. By contrast, other works have shown that adapting the initial layers (i.e. bottom-adapter) can be important for transferring across large domain-shifts (i.e. sim2real (Jeong

Figure 3: Different locations to insert adapter modules for convolution (Left) and transformer (Right) models.

et al., 2020), images to line-drawings (Aytar et al., 2017)). One can also inject mid-level adapters as demonstrated in many non-control settings (Rebuffi et al., 2017; Houlsby et al., 2019).

### 3.1 ADAPTER MODULES

Adapter modules are light-weight neural modules that can be inserted at different layers of a pretrained deep neural network. Prior works have explored adapter modules for transfer learning, wherein adapter modules are inserted at each layer of a pretrained deep network and only these adapters are updated at the fine-tuning stage (Houlsby et al., 2019). Overall, adapter modules have two important properties, 1) they are *lightweight*, i.e., they have fewer parameters compared to the original network, and 2) they keep the *initialization* provided by the pretrained deep network. Below, we provide a short discussion on how adapter modules achieve these properties.

Prior works limit the number of parameters in the adapter modules by either using $1 \times 1$ convolutions (Rebuffi et al., 2017) or via bottleneck architectures for transformers (Houlsby et al., 2019). In our formulation, we use a bottleneck architecture for both convolution- and attention-based architectures, as well as utilizing $1 \times 1$ convolutions for the former. Specifically, for a given input $x \in \mathbb{R}^{n \times f}$ where $n$ is the number of samples, $f$ is the input feature dimensionality, adapter outputs are parameterized as a combination of down- (d) and up-projection (u) weights: $x' \leftarrow W_A^u \left( h \left( W_A^d x \right) \right)$, where $h$ is some non-linearity, $W_A^d \in \mathbb{R}^{f \times f'}$ and $W_A^u \in \mathbb{R}^{f' \times f}$ are the adapter ($A$) weights, $f'$ is the bottleneck feature size. Since $f' << f$, the adapter modules utilize a very small number of parameters in comparison to the original pretrained network. Importantly, we use the above formulation for both convolutional and transformer based architectures – for convolutional architectures we downproject across the channel (feature) dimension, while for transformer based architectures we downproject along the feature dimension for each patch. Figure 2 visualizes the adapters used in our work.

To fully utilize the initialization provided by the pretrained weights, we initialize the adapter modules with weights close to 0 and add skip connections when inserting the adapter modules into the pretrained network. This ensures that adapter modules have no effect at initialization and thus the pretrained network's initialization remains intact. Overall, for an input $x$, the output for a pretrained model's layer with an adapter is $x' \leftarrow g_{\text{pretrain}}(x) + g_{\text{adapter}}(x)$. Also note that the adapter modules can also be added serially, i.e., directly after the output of a pretrained layer, in which case $g_{\text{pretrain}}$ can be viewed as an identity function. Finally, similar to previous works (De Vries et al., 2017; Perez et al., 2018; Houlsby et al., 2019) we also add offsets to the normalization parameters used in different architectures (e.g. layernorm (Ba et al., 2016) for vision transformers).

### 3.2 VISUAL ADAPTERS FOR CONTROL

Prior works often utilize adapter modules for a broadly similar set of tasks in a given domain. For instance, text classification tasks (Houlsby et al., 2019), neural machine translation tasks (Pham et al., 2020), or few-shot visual classification tasks (Li et al., 2022). Additionally, the pretrained model used in these tasks (e.g. masked language modeling (Devlin et al., 2018) or ImageNet pretraining) is strongly correlated with the downstream tasks. By contrast, there exists much larger *task differences* between visual pretraining tasks (e.g. image-inpainting/classification) and downstream robot manipulation (continuous action prediction). While visual classification tasks mostly require semantic features, manipulation tasks require actionable features ("what" and "where" the objects are to predict actions) (Dwibedi et al., 2018; Rosinol et al., 2020). These actionable features need both semantic and spatial features.

These task differences manifest in a large domain shift between visual pretraining tasks and downstream robot manipulation. For instance, robot manipulation data consists of table-top settings, moving robot arm, heavy occlusion, which is very different from object-centered images of Im-

Figure 4: Different environments we evaluate our approach on. For Metaworld and Kitchen suites we frollow the setup from (Nair et al., 2022) including the same set of demonstrations. For RGB-Stacking suite we use Skill Mastery setting (Lee et al., 2021).

ageNet. Based on above differences we next discuss how and where to efficiently insert adapter modules to improve upon "off-the-shelf" representations produced by the fixed pretrained network.

While we can add adapter modules through all layers of the pretrained network, such a choice is highly parameter inefficient and redundant especially for large networks with many layers. We coarsely categorize the network layers based on their functional forms as *bottom*, *middle* and *top* sections as visualized in Figure 3. Next, motivated by the above discussion on visual adapters for control, we provide intuitive reasoning for injecting adapters in each section. Importantly, as we show empirically, using adapters at each of these network sections is important for task performance.

The *bottom* layer directly uses the raw images as input. In scenarios, where there is a mismatch between the downstream task's image observations and the pretrained bottom layer feature statistics, the downstream task performance can be sub-optimal. As discussed above, such scenarios are common for downstream manipulation tasks, since there exists a significant domain gap between the data distribution of pretrained vision models (often in-the-wild data) and standard table-top settings with much closer and non-canonical camera views.

The *middle* category, which contains most of the fixed pretrained network ($\approx 90\%$) weights, is used to extract the appropriate input abstraction. However, these network weights are trained on visual learning tasks which often focus on semantic understanding (e.g. image classification) instead of spatial and causal understanding which are important for control. Nevertheless, earlier layers of the network are known to capture useful invariances (Zeiler & Fergus, 2014; Parisi et al., 2022). Hence, *sparsely* inserting adapter layers through the pretrained network can allow us to better adapt "off-the-shelf" representations for downstream manipulation tasks. Finally, we note that the output of the middle category is a set of spatial features for convolutional networks and patch features for ViTs.

The *top* category uses the spatial representation from the middle category as input and outputs the robot action. This high dimensional spatial representation (size $\approx 20K$) is converted into a smaller representation (size $\approx 2K$) either via average/max pooling or by down-projecting using $1 \times 1$ convolutions or a small shared MLP. Finally, this smaller representation can be used to directly output the action using a linear policy head. While a linear head is sufficient in settings where there is strong task alignment (Chen et al., 2020), given the large difference in the pretraining and downstream tasks, additional top layer adaptation helps further improve performance. We note that most prior works that use fixed pretrained features (Pari et al., 2021; Nair et al., 2022) also utilize such top layer adaptation. We discuss implementation details for specific architectures in Section 4.2.

**Training Adapters for Control:** We use behaviour cloning to learn task policies from demonstrations. We use euclidean loss for action prediction to optimize adapter parameters.

## 4 EXPERIMENTAL SETUP

We evaluate our use of adapters for large pretrained visual models across three different environment suites each with increasing task complexity and across three different network architectures.

### 4.1 MANIPULATION TASKS

In this work, we consider Metaworld (Yu et al., 2020), Franka-Kitchen (Gupta et al., 2019), and RGB-Stacking task suites (Lee et al., 2021). Figure 4 visualizes the tasks considered from each of these task suites. Both Metaworld and Kitchen have been used previously (Nair et al., 2022)

to evaluate fixed "off-the-shelf" pretrained visual representations. Hence, for both suites we use the same environments and demonstrations. Overall, we use 5 different environments from the Metaworld and Kitchen task suites. For each environment we use 3 different camera configurations as provided by Nair et al. (2022). Additionally, similar to previous work we use 25 demonstrations for each environment and train a separate policy for each environment and camera configuration.

While both Metaworld and Kitchen suites consider many tasks, each task often has a very narrow state-space distribution. For instance, Kitchen tasks use fixed object positions while MetaWorld tasks have limited position variance only. Hence, we also evaluate our approach on a much more challenging RGB-stacking suite (Lee et al., 2021) (Figure 4 bottom). The RGB-stacking tasks involve three geometric objects colored red, green, and blue and the goal is to stack the red object on top of blue object. Object geometries should also be taken into account for a successful stacking behaviour. These tasks randomize object positions and orientations in a basket and thus result in a large initial state-space distribution. Furthermore, as these tasks consider a diverse range of shapes, stacking requires precise control and cannot be achieved simply by dropping one object onto the other. To obtain demonstrations for these tasks, we initially train an RL policy and collect 100,000 demonstrations from it. We use the *Skill Mastery* setting from (Lee et al., 2021) for evaluation.

## 4.2 NETWORK ARCHITECTURES

We evaluate the effectiveness of adapters for manipulation tasks in the context of three different network architectures. Specifically, we use normalizer-free networks (NFNet) (Brock et al., 2021), residual networks (ResNet) (He et al., 2016) and vision transformers (ViT) (Dosovitskiy et al., 2020). Among the different architectures within each category we use *NFNet-f0*, *ResNet-50* and *ViT-B/16*. In addition to imagenet pretraining across all three architectures, we also evaluate using ALIGN (Jia et al., 2021) for NFNet, BYOL for ResNet (Grill et al., 2020) and masked auto-encoder (MAE) for ViT (He et al., 2022). For fair comparison with prior works (Nair et al., 2022) we avoid using any data augmentations during training.

**Adapter Parameterizations:** As noted previously in Sub-Section 3.2 we use *bottom*, *middle* and *top* adapters. We discuss the specific parameterizations for each of these adapter types. Table 4 provides an overall adapter parameter count. We provide detailed parameterization in Appendix A.1.

*Bottom:* For NFNet and ResNet architectures we use the initial convolution, while for ViT we use the initial patch embedding as the only bottom set of layers. We add 1 adapter for this bottom layer.

*Middle:* For middle adapters we use 4 and 6 adapters for the convnet and transformer based architectures respectively. For the convnet based architectures we add each adapter to the first convolution in each block group except the last group, where we add it at the end. While for the transformer based architectures we apply them at layers $\{0, 1, 5, 6, 10, 11\}$.

*Top:* We project middle layer's spatial features onto a lower dimensional space. To process these features we *optionally* add 2 layer MLP with 256 dimensions, we refer to these as top adapters.

**Evaluation:** We evaluate our approach using 40 rollouts from the BC learned policy. We use mean success rate of the final policy as our metric. When providing task suite metric we average the mean success rate across all environments and camera configurations. We also note that some previous works evaluate the BC policy at fixed training step intervals and use the maximum over the mean success rates at each interval as the metric. However, using max as a statistic is not robust and is easily influenced by outliers. While comparing with previous works we use the max statistic, however for all other results we use the mean. Training details and hyper-parameters are in Appendix A.

## 5 RESULTS

With our experiments we aim to show: 1) Using fixed pretrained representations often leads to suboptimal task performance as compared to directly adapting the representation for the downstream task, especially for more challenging manipulation tasks. 2) Inserting small adapter modules into a fixed pretrained network (i.e. lossless adaptation) is sufficient to reach full-finetuning performance across a wide range of manipulation tasks and network architectures. 3) Adapter modules are parameter efficient and robust to different initializations of the fixed large pretrained vision model (e.g. via different visual pretraining strategies). We look at each point in the following subsections.

| | Metaworld | Franka-Kitchen | RGB Stacking |
|---|---|---|---|
| Fixed Pretrained Feat. - ImNet (R3M) (Nair et al., 2022) | 0.66 | 0.30 | - |
| Fixed Pretrained Feat. - MoCo-345 (R3M) (Parisi et al., 2022) | 0.64 | 0.38 | - |
| Fixed Pretrained Feat. - R3M (R3M) (Nair et al., 2022) | 0.78 | 0.56 | 0.30 |
| Fixed Pretrained Feat- (Ours - ImNet) | 0.62 | 0.48 | 0.28 |
| Full Finetune (Ours) | 0.98 | 0.72 | 0.70 |
| Fixed Pretrained Feat. - (Ours ImNet - mean succ) | 0.44 | 0.34 | 0.14 |
| Full Finetune (Ours - mean succ) | 0.91 | 0.57 | 0.49 |

Table 1: Success rate comparison using fixed pretrained features (using pretrained large vision models) with methods that update the pretrained visual features on the downstream manipulation task.

| | | Metaworld | | | Franka-Kitchen | | | RGB Stacking | |
|---|---|---|---|---|---|---|---|---|---|
| | Pretrain Feat. | Adapters | Full FT. | Pretrain Feat. | Adapters | Full FT. | Pretrain Feat. | Adapters | Full FT. |
| NFNet | 0.44 | 0.82 | 0.94 | 0.12 | 0.39 | 0.42 | 0.14 | 0.45 | 0.47 |
| ResNet | 0.39 | 0.80 | 0.88 | 0.12 | 0.24 | 0.25 | 0.14 | 0.48 | 0.49 |
| ViT | 0.19 | 0.78 | 0.84 | 0.15 | 0.26 | 0.25 | 0.18 | 0.49 | 0.48 |

Table 2: Mean success rate comparisons between using *fixed* pretrained features, adapters and full fine-tuning across all three different environments with three different architecture choices.

## 5.1 FIXED PRETRAINED FEATURES VS ADAPTER REPRESENTATIONS

**Fixed Off-the-Shelf Representations:** In the first part of our experiments we show that while fixed off-the-shelf representations (without any adaptation) are useful, they can be highly sub-optimal for the given downstream task. To show this we compare the fixed representations extracted using pretrained weights (Pretrained Feat.) obtained via supervised imagenet pretraining and compare them with full fine-tuning (Full FT). Table 1 compares these across all task suites. For a fixed comparison with previous works Table 1 reports results for the ResNet-50 model, since previous works only evaluate the ResNet architecture. As seen in Table 1, fixed off the shelf-representations are comparatively much worse across all environment suites. For Metaworld, Kitchen and RGB-Stacking suites the *relative* change in performance is around $\approx 20\%, 30\%$ and $100\%$ respectively. Also, for the RGB-stacking suite the mean performance is much lower $14\%$, which shows that fixed pretrained representations become significantly less effective for challenging manipulation tasks.

**Lossless Adaptation of pretrained Visual Features:** We now show that our proposed adapters can match full-fine-tuning performance for downstream manipulation tasks without losing any existing information. Table 2 compare full fine-tuning (Full FT.) approaches with our adapters (Adapters) as well as fixed pretrained features (Pretrained Feat.) For these and future results we report metrics using the more robust mean statistic. Additionally, for task suites with limited state-space distributions, *i.e.*, Metaworld and Franka-Kitchen, we avoid using any proprioceptive information (see Appendix B for results with proprioceptive). This allows us to robustly verify the visual representation and avoids any proprioceptive information leakage which can allow the robot to solve the task even without using the visual features.

Table 2 shows the results for each task suite and network architecture combination. For the adapter results we report results with bottom, middle and top adapters. As before, for a fair comparison we use top adapters for all other approaches. As seen in the above table, our parameter efficient adapters can closely match full fine-tuning performance across all environment suites and architectures. Most noticeably, for both Franka-Kitchen and RGB-Stacking tasks, our use of adapters is able to exactly match (average difference in performance $< 2\%$) the performance derived from full fine-tuning (Full FT). While there exists a slightly larger gap for the metaworld environments – average performance difference $\approx 6\%$. However, compared with directly utilizing the fixed pretrained features, we see a huge performance increase of around $30\%$ averaged over all tasks and architectures. In addition to the average results, Table 2 also shows that there exists a performance increase across all architectures. Our results clearly show that the features derived from visual pretraining are quite useful and inserting few additional adapters allows us to achieve close to optimal performance without sacrificing any of the previously learned abilities of the pretrained vision models.

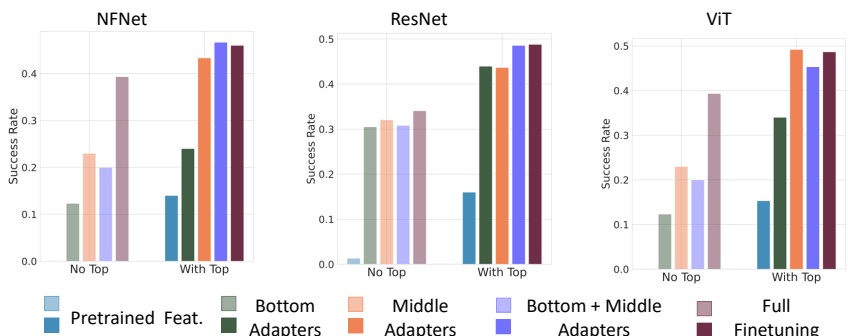

Figure 5: Ablation results on the RGB-Stacking environment for 3 different network architectures.

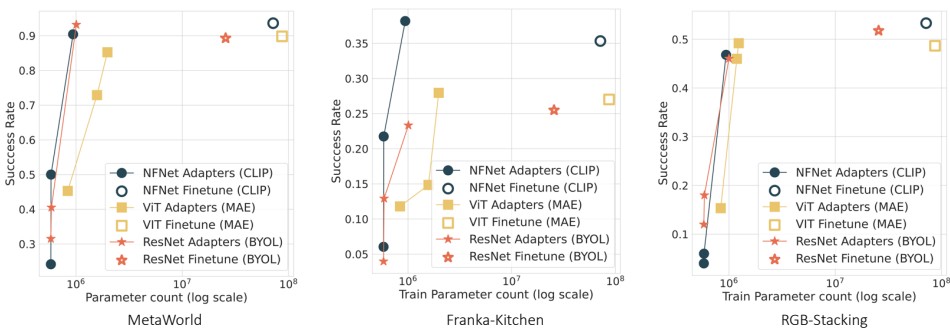

Figure 6: Results with different pretraining initializations (for 3 different models) all 3 environments – NFNet with CLIP, ResNet with BYOL and ViT with MAE. *Bottom Left:* points plots performance of fixed pretrained features with top adapters. *Top Right:* points plot full fine-tuning performance (with top adapters). *Solid Lines:* indicate adapter performance with adapters first added to bottom and then middle layers.

## 5.2 Effects of Adapter Locations & Different Pretrained Representations

We investigate the effect of inserting adapters in each of the different network layers (see Section 3.2). Figure 5 shows results for inserting adapters in each network layer for RGB stacking suite (results and discussion for all suites are provided in Appendix B.2). We split each network plot above into two parts – 1) without using top layer adapters (i.e. directly using a linear policy head), and 2) using a top layer adapter (i.e. using 2 layer MLPs before the linear policy head).

From Figure 5 we see that *without* top layer adapters (greyed out left plots) the performance for all methods decreases – more so for fixed pretrained features. As seen above, top adapters are crucial, e.g., not using top layer adapters almost *halves* the control performance across all different architectures. Figure 5 also shows that bottom adapters alone can give a significant performance boost (almost $2\times$ than Pretrained Feat. with top adapters). Since bottom adapters for conv-net architectures have very few parameters (a few thousands, see Table-4), this performance increase is not merely a function of more parameters and thus shows the need for better adaptation of pretrained visual features. Overall, best performance is often achieved when using all top, middle and bottom adapters together. Since previous works (Rebuffi et al., 2017; Li et al., 2022) only consider adapters in middle sections, our results show the benefits of bottom/top adapters for control tasks.

**Adapters with Different Pretrained Representations:** We now show that our proposed adapters give similar benefits with pretrained weights obtained from vastly different pretraining (pretext) tasks. For NFNet we use CLIP pretraining (Radford et al., 2021), for ResNet we use BYOL (Grill et al., 2020) and for ViT we use masked auto-encoder (MAE) (He et al., 2022). Figure 6 plots the result for each of these architectures across all three task suites. The X-axes shows the number of train parameters (see Table 4). The *bottom left* points in each of the above plots indicate the performance of fixed pretrained features. While *top right* points show full fine-tuning's performance. On the other hand, the solid lines indicate the performance improvements on inserting bottom and middle adapters (top adapters are used for all approaches). As seen in Figure 6, for all tasks and

|  | Triplet 1 | Triplet 2 | Triplet 3 | Triplet 4 | Triplet 5 | Mean |
|---|---|---|---|---|---|---|
| NFNet - Pretrained Feat. | 0 | 0 | 0 | 0 | 0 | 0.0 |
| NFNet - Scratch | 0.02 | 0.02 | 0.01 | 0.16 | 0.04 | 0.05 |
| NFNet - Adapters | 0.18 | 0.14 | 0.16 | 0.47 | 0.23 | 0.24 |
| NFNet - Full FT. | 0.22 | 0.36 | 0.15 | 0.76 | 0.26 | 0.35 |
| NFNet - Pretrained Feat. (DR) | 0.00 | 0.02 | 0.0 | 0.06 | 0.03 | 0.02 |
| NFNet - Adapters. (DR) | 0.38 | 0.40 | 0.38 | 0.88 | 0.64 | **0.53** |
| NFNet - Full FT. (DR) | 0.40 | 0.36 | 0.32 | 0.91 | 0.62 | **0.52** |
| ViT - Adapters | 0.04 | 0.08 | 0.04 | 0.24 | 0.12 | 0.10 |
| ViT - Full FT. | 0 | 0 | 0 | 0 | 0 | 0 |

Table 3: Sim2Real results for RGB-Stacking Task with and without using any visual domain randomized (DR) data for learning the manipulation task policy.

|  | Bottom | Middle | Top | Full |
|---|---|---|---|---|
| NFNet-f0 | 0.5K | 0.4M | 0.6M | 71M |
| ResNet-50 | 6.9K | 0.4M | 0.6M | 25M |
| ViT | 0.74M | 0.4M | 0.9M | 86M |

Table 4: Number of parameters to be learned for different adapters as well as full fine-tuning.

pretrained weights adapters are able to closely match the performance of full fine-tuning approach, while only registering a very marginal increase in the number of trainable parameters.

Additionally, comparing MetaWorld results in Figure 6 (Left) and Table 2 we see that while there exists a minor gap between adapters and full-FT with imagenet-supervised weights $\approx 6\%$, this gap reduces significantly for self-supervised pretraining weights. This advantage of MAE features for control is also observed in (Xiao et al., 2022). Additionally, similar to Radosavovic et al. (2022), we also find that CLIP pretrained weights (with top adapters only) can perform poorly. For instance, they get $< 5\%$ on RGB-stacking tasks. However, using adapters the performance significantly improves $\approx 50\%$ and closely matches full fine-tuning performance. Moreover, the adapted representations match the performance of more performant models (e.g. MAE).

## 5.3 SIM2REAL RESULTS

Finally, we investigate if large scale visual pretraining combined with our use of adapters can allow for *sim2real transfer*. Prior works that utilize fixed pretrained vision models for real robot tasks often only evaluate on tasks requiring simple motions (reach/grasp) and almost always train the policy on real robot data (Shridhar et al., 2022; Nair et al., 2022). By contrast, we show results for sim2real transfer, i.e, we use no extra real-world robot data. We use the challenging RGB-stacking suite for evaluation, and evaluate sim2real transfer both *with* and *without* visual domain-randomization data. Results are reported with 100 rollouts for each policy and object triplet.

Table 3 shows results for one convolutional (NFNet) and one transformer (ViT) architecture. We find that ViT based policies perform poorly compared to NFNet policies in this setting, and a fully fine-tuned ViT policy is completely unable to solve any task. We suspect this may be due to the high capacity of transformer-based models which allows them to quickly overfit on new data (Wortsman et al., 2022).Interestingly, using adapters instead of fine-tuning partially mitigates this issue. While the average performance is poor $10\%$, it is able to achieve $24\%$ success on the easier setting (triplet 4). Table 3 further shows the impressive performance of NFNet based policies. For full fine-tuning and adapter approaches NFNet policies can achieve $35\%$ and $24\%$ success rate, while training from scratch only achieves $5\%$ and fixed pretrained features do not result in any successes. Finally, we also evaluate the NFNet policies with visual domain randomization data (DR rows). These policies show the strongest performance $\approx 53\%$, matching performance in simulation. Overall these results corroborate the well-known benefits of DR for sim2real, but also suggest that adding pretrained vision models can improve performance significantly over learning-from-scratch.

## 6 CONCLUSION

In this work we propose the lossless-adaptation problem, i.e., we aim to adapt representations from large-scale pretrained vision models for close to optimal manipulation task performance. We show that using fixed representations by solely using top-adapters (as is common) can fail to achieve optimal task performance especially for challenging manipulation tasks. To solve this we propose our parameter efficient adapters. We show that that inserting these adapters at appropriate network locations can achieve close to optimal downstream task performance (closely matching full fine-tuning performance). We show that our results hold across 3 different manipulation suites, 3 different network architectures and multiple different pretrained weights. Further, we show that adapters also allow for sim2real transfer, all while maintaining the pretrained network's original capabilities.

## 7 ETHICS STATEMENT

We have considered possible concerns regarding the ethics of this line of work, and to the best of our knowledge, can recognize no major reasons for concern. Our work uses pre-existing pre-training datasets that are publicly available, and that, to the best of our knowledge, are filtered to excise personally identifiable human data. This work does not delve into applications of models that might adversely affect or cause discrimination against groups. Our methodologies, as far as we can tell, would not be immediately translatable to harmful applications. The work has been screened internally (a technical and strategy check by non-author parties) for potential privacy and security issues.

## 8 ACKNOWLEDGMENTS

We thank Mel Vecerik, Oleg Sushkov, Yi Yang, Todor Davchev, Akhil Raju and other members of the DeepMind Robotics team for their useful insights, helpful discussions and infrastructure support throughout the course of this project.

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

## A    EXPERIMENTAL SETUP

We provide further training details for our approaches in the following sections. First we discuss the adapter parameterizations in more detail specifically detailing the adapter locations and their associated trainable parameters.

### A.1    ADAPTER PARAMETERIZATIONS

As noted previously in Sub-Section 3.2 we use *bottom*, *middle* and *top* adapters. We discuss the overall number of parameters for each of these adapter types. Table 4 provides an overall count for the number of adapter parameters.

*Bottom:* For NFNet and ResNet architectures we use the initial convolution, while for ViT we use the initial patch embedding as the only bottom set of layers. We add 1 adapter for this bottom layer. Since the image input only contains 3 channels and the bottom layer consists of only 16 and 256 channels for NFNet and ResNets respectively, bottom adapters require very few ($< 1000$) parameters. While for ViT since we have a large feature output, it results in a much larger number of parameters $\approx 0.7M$.

*Middle:* For middle adapters we use 4 and 6 adapters for the convnet and transformer based architectures respectively. For the convnet based architectures we add each adapter to the first convolution in each block group except the last group, where we add it at the end. While for the transformer based architectures we apply them at layers $\{0, 1, 5, 6, 10, 11\}$ which yields around $\sim 400K$ adapter parameters. While better choices and adapter placements may exist we use these uniformly across all tasks. Finally, the output of middle layer consists of a set of spatial features. For NFNet and ResNet these are output of layer 4 and final conv with a spatial size of $7 \times 7$ and a feature size of 3096 and 2048 respectively. For ViTs we get 196 patches each with 768 features.

*Top:* As noted previously in Sub-section 3.2, the high dimensional spatial features from the middle layer can be reduced either via mean pooling or by projecting them onto a lower dimensional space through a single layer. Since prior works use mean pooling we use it to compare our work with them. However, since manipulation tasks are spatial in nature we also investigate down-projecting the high dimensional spatial features into smaller dimensions and concatenating them. This formulation avoids any loss of spatial information. To achieve this for NFNet and ResNets we use 1 convolution layer with $1 \times 1$ kernel and 41 output channels. While for ViT we use a shared MLP that projects each patch embedding into a 20 dimensional feature. Finally, to further process these features we *optionally* add 2 MLPs each with 256 parameters, we refer to these as top layer adapters.

*Policy Head:* We universally use a linear policy head that converts the output of the top layer into the robot action to be executed.

### A.2    TRAINING DETAILS

As noted in the main paper we use behavior cloning with mean squared loss as the optimization objective. We use a linear policy head to predict continuous actions. While specific action parameterizations, such as the use of binary gripper can help increase performance, for a fair comparison of representations we avoid using any such techniques. Table 5 lists out the detailed hyperparameters used in our experiments. We uniformly use the same set of hyperparameters across most task settings except the learning rate, wherein we found using a slightly higher learning rate of $1e - 3$ works better for Franka-Kitchen tasks.

*Network Details*: As discussed before our implementation uses three different network architectures – NFNets, ResNets and ViTs. Figure 3 presents the overall architecture. In settings where we use proprioceptive information, we use a single linear layer with 256 dimensions to map the low dimensional proprioceptive information to a higher dimensional space. This high dimensional proprioceptive information is then concatenated with the visual features before being forwarded to the 2 layer MLP (each with 256 units). Further, since we evaluate our approach across very different architectures we use the same policy form across all of them. Thus, we avoid using any normalization techniques such as BatchNorm or LayerNorm in our policy implementation.

| Training Parameters | MetaWorld | Franka-Kitchen | RGB Stacking |
|---|---|---|---|
| Loss | MSE | MSE | MSE |
| Optimizer | Adam | Adam | Adam |
| Learning Rate | 1e-4 | 1e-3 | 1e-4 |
| Weight Decay | 1e-6 | 1e-6 | 1e-6 |
| Gradient Norm Clip | 1.0 | 1.0 | 1.0 |
| Training Steps | 40K | 40K | 200K |
| Learning Rate Schedule | cosine | cosine | cosine |
| Learning Rate Schedule Warmup Steps | 5K | 5K | 10K |
| Adapter Features Size | 32 | 32 | 32 |

Table 5: Training Details for each of the three different task suites used in our work. For each task within the task suite we use the same set of hyperparameters.

| | Assembly | Bin-Picking | Button Press | Drawer Open | Hammer | Average |
|---|---|---|---|---|---|---|
| NFNet | 0.92 | 0.7 | 0.94 | 0.96 | 0.94 | 0.89 |
| ResNet | 0.90 | 0.66 | 0.96 | 0.94 | 0.92 | 0.88 |
| ViT | 0.92 | 0.8 | 0.91 | 0.98 | 0.92 | 0.91 |

Table 6: Task specific results for using bottom, middle and top adapters with proprioceptive information (proprio) for each task in MetaWorld.

# B  ADDITIONAL RESULTS

We provide further results for the use of adapters in the three different environment suites considered in the main paper. We then discuss the ablation results on adapter locations for all suites and network architectures. For these results, in addition to average metrics across all enviroments, we also provide task specific metrics.

## B.1  ADAPTER RESULTS WITH PROPRIOCEPTIVE INFORMATION

In this section we present detailed task-specific success rate using our proposed adapters for each task in the three manipulation suites. For these results we use all top, bottom and middle adapters in our implementation. Further, in addition to visual features we also utilize proprioceptive information for these results. Table 6, 7 and Table 8 report task-specific results for MetaWorld, Franka-Kitchen and RGB Stacking suites using all three different architectures with imagenet pretrained weights. Comparing Table 6 with previous results in Table 2 we see that adding proprioceptive information results in $\approx 10\%$ increase in the average success rate. This increase holds consistently across all architectures. More interestingly we also find that for most tasks (except Bin-Picking) the agent can reach greater than $90\%$ performance, while for some tasks such as *Button-Press* and *Drawer-Open* it can even reach close to 100% performance.

Table 7 shows the results for each task in the Franka-Kitchen suite. Compared to previous results in Table 2 we see a much larger increase in the performance ($\approx 60\%$ relative performance increase on average) of each architecture in the Franka-Kitchen suite. One reason for such a large increase is the very limited state space distribution for these tasks. Since all objects in the environment are fixed and only the initial robot configuration changes, it is much easier for the robot to memorize the proprioceptive information and map it to observed expert actions for improved task performance. Additionally, while both MetaWorld and Franka-Kitchen use 25 demonstrations, each demonstration in MetaWorld has 500 steps while in Franka-Kitchen each demonstration is only 50 steps. This results in $10\times$ difference in the amount of training data. However, since prior works use these settings for a fair comparison we follow similar evaluation protocols.

## B.2  EFFECTS OF ADAPTER LOCATIONS

In this section we investigate the effect of inserting adapters in each of the different network layers as discussed in Subsection 3.2 and initially explored in Subsection 5.2. Due to space constraints in Subsection 5.2 we only provide results for the RGB Stacking task. In this subsection we show

|        | Knob1-On | LDoor-Open | Light-On | Micro-Open | SDoor-Open | Average |
|--------|----------|------------|----------|------------|------------|---------|
| NFNet  | 0.46     | 0.44       | 0.72     | 0.32       | 0.94       | 0.58    |
| ResNet | 0.48     | 0.46       | 0.60     | 0.30       | 0.88       | 0.54    |
| ViT    | 0.6      | 0.48       | 0.59     | 0.36       | 0.83       | 0.57    |

Table 7: Task specific results for using bottom, middle and top adapters with proprioceptive information (proprio) for each task in Franka-Kitchen suite.

results across all manipulation suites and network architectures. For ease of comparison we also plot the RGB Stacking results from before.

Figure 7 shows results for inserting adapters in each network layer across all 3 task suites and architectures. As noted before, we split the results in each plot into two parts. 1) without using top layer adapters (i.e. directly using a linear policy head), and 2) using a top layer adapter (i.e. using 2 additional MLPs before the linear policy head). Moreover, in addition to the different adapter locations we also show results for fixed pretrained representations (Pretrain Feat.) and full fine-tuning (Full FT.) both with and without top adapters. As noted in the main paper, prior works always use such top adapters in their implementations.

*Top Adapters:* In our discussion in Subsection 5.2 we showed that for the RGB Stacking task top adapters are quite important to achieve close to optimal task performance. We note that the greyed out plots in Figure 7 indicate methods that do not use top adapters. Our results in Figure 7 show that this holds true for both MetaWorld and Franka-Kitchen suites as well. For both of these suites we find that using top adapters improves the downstream manipulation performance. However, as seen in the metaworld results (top row of Figure 7), full fine-tuning approaches (last bar in each plot) can reach good performance even without top adapters. However, this does not hold for the Franka-Kitchen tasks (middle row in Figure 7). We hypothesize this is because of the metaworld setup, wherein there is usually a single object centered on an otherwise empty table, which presents an easier visual setting and simply fine-tuning the high capacity pretrained visual model can extract the appropriate task representation. However, we do note that our use of adapters is able to closely match the full fine-tuning performance across all architectures.

*Bottom Adapters:* Similar to RGB-stacking results before we note that the bottom adapters (plotted in Green) with very few parameters (around a few thousand) can lead to substantially better results than simply using fixed pretrained models. This holds when bottom adapters are combined with top adapters and even in the absence of top adapters. Although, as noted before the overall results are much poorer without top adapters. From Figure 7 we see that bottom adapters help for both NFNet (green bar in column 1, row 1 and column 1, row2) and ResNet (green bar in column 1, row 1 and column 1, row2). Thus, broadly similar results hold across environment suites.

*Middle Adapters:* From Figure 7 also shows that while bottom and top adapters *together* (green bar on the right plots) can achieve good performance there still exists a significant gap compared to the full fine-tuning approach. However, inserting middle adapters, either alone (shown by orange) or together with bottom adapters (shown in purple) leads to a much more improved performance. Overall, using adapters in all the layers is closely able to match the full fine-tuning performance. This substantial effect of middle adapters is not unexpected since the middle part of the network contains a large part of the pretrained network and thus has significant affect on the output representation.

Overall our results show the importance of adding adapters in each of the network section. As noted previously, this usage of adapters in different network sections is different from prior works (Rebuffi et al., 2017; Li et al., 2022) which only focus on middle adapters. While such middle adapters are sufficient for semantic classification tasks (considered in (Rebuffi et al., 2017; Li et al., 2022), they are insufficient for control tasks considered here.

## C  DISCUSSION

In this section we briefly discuss some observations on the use of adapters and future works that adapters can enable for robot control tasks.

|         | Triplet 1 | Triplet 2 | Triplet 3 | Triplet 4 | Triplet 5 | Average |
|---------|-----------|-----------|-----------|-----------|-----------|---------|
| NFNet   | 0.40      | 0.18      | 0.11      | 0.67      | 0.90      | 0.45    |
| ResNet  | 0.48      | 0.34      | 0.11      | 0.62      | 0.86      | 0.48    |
| ViT     | 0.25      | 0.40      | 0.13      | 0.80      | 0.85      | 0.49    |

Table 8: Task specific results for using bottom, middle and top adapters for each task in RGB-Stacking suite.

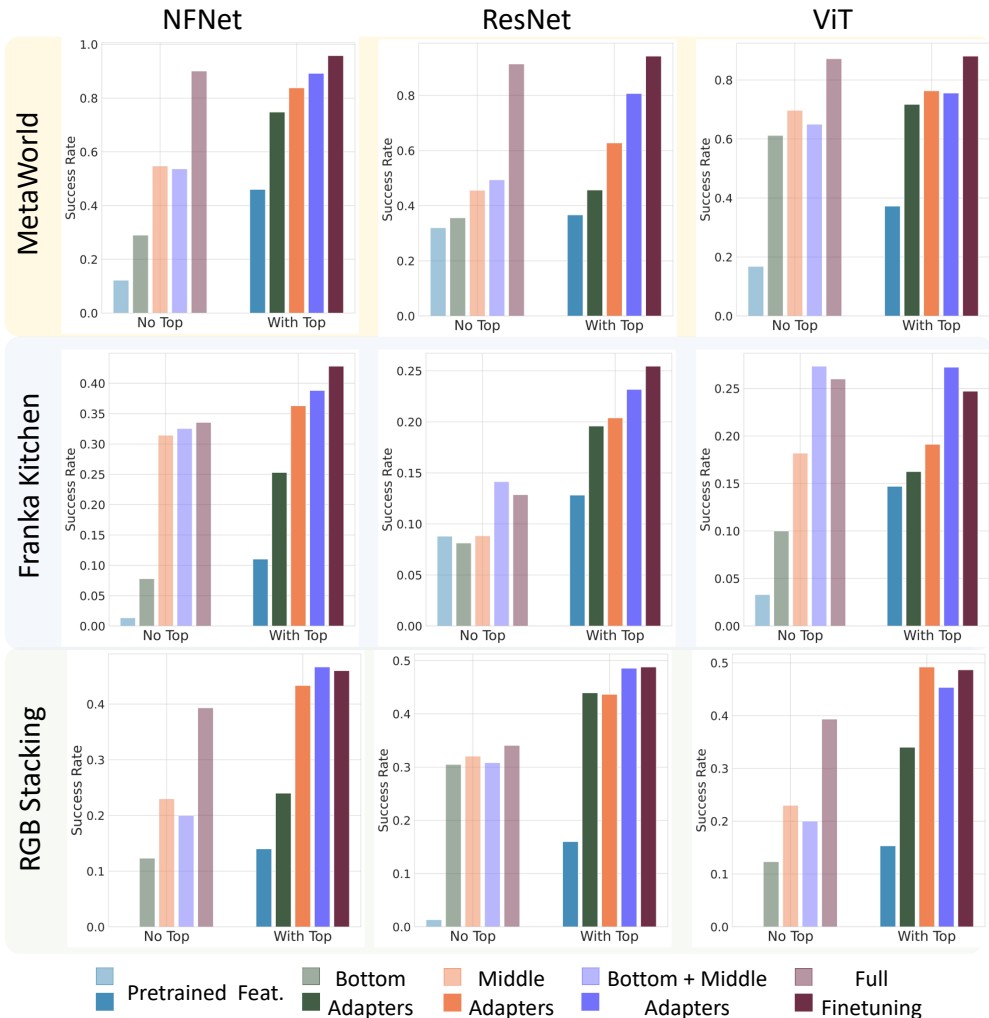

Figure 7: Results on the RGB-Stacking environment for 3 different type of model architectures.

*Sim2Real:* Prior work in robotics often use extensive visual domain randomization, photo-realistic simulators, and other sensory modalities such as depth (which have a smaller sim2real gap) to transfer simulation trained policies to real-world. Our results show that using large pre-trained models (which are closer to natural image statistics) can avoid the need for such expensive sim2real strategy. While there still exists a performance gap ($\sim 24\%$ without visual domain randomization, $\sim 53\%$ with domain randomization), we believe this opens up significant avenues for future research.

*CLIP/ALIGN Representations:* Recent works (Nair et al., 2022; Radosavovic et al., 2022) propose representations that outperform off-the-shelf CLIP representations. However, CLIP representations are much more semantically powerful (Gadre et al., 2022) and highly robust across a range of data distributions (Wortsman et al., 2022). Our results in Section 5.2 shows that while off-the-shelf

CLIP representations can be poor (especially for RGB-stacking see Figure 6 Right), adapting them through our proposed adapters results in similar performance as other adapted representations (such as MAE ones). Moreover, adapting CLIP representations significantly outperforms all fixed off-the-shelf representations. This is important since CLIP representations are much more semantically powerful and have been used for vastly different task – dialogue generation (Alayrac et al., 2022), robot navigation (Gadre et al., 2022), images sketching (Vinker et al., 2022). Our result shows that the same pretrained model can perform robot manipulation and reach close to optimal performance. This provides exciting avenues for future research wherein a single fixed model can be used to solve a very wide range of tasks.

