# OpenReview forum: "Lossless Adaptation of Pretrained Vision Models For Robotic Manipulation"
_ICLR.cc/2023/Conference — ICLR 2023 poster_

### Official Review · Reviewer_a8EG · 2022-10-23

**Confidence:** 4
**Correctness:** 3
**Technical Novelty And Significance:** 2
**Empirical Novelty And Significance:** 3
**Recommendation:** 6

**Clarity, Quality, Novelty And Reproducibility:**

### Clarity and Quality
The paper reads well, but Figure 3's caption is a bit difficult to parse without reading the main draft.

### Novelty
As far as I'm aware, introducing adapters is already proposed in the context of natural language processing [1, 2, 3, 4], as also discussed by the authors in Related Work. Hence the main novelty of the paper is not in the method. It would be interesting to consider the approach similar to [5].

[1] Houlsby, Neil, Andrei Giurgiu, Stanislaw Jastrzebski, Bruna Morrone, Quentin De Laroussilhe, Andrea Gesmundo, Mona Attariyan, and Sylvain Gelly. "Parameter-efficient transfer learning for NLP." In International Conference on Machine Learning, pp. 2790-2799. PMLR, 2019.

[2] Karimi Mahabadi, Rabeeh, James Henderson, and Sebastian Ruder. "Compacter: Efficient low-rank hypercomplex adapter layers." Advances in Neural Information Processing Systems 34 (2021): 1022-1035.

[3] Mahabadi, Rabeeh Karimi, Sebastian Ruder, Mostafa Dehghani, and James Henderson. "Parameter-efficient multi-task fine-tuning for transformers via shared hypernetworks." arXiv preprint arXiv:2106.04489 (2021).

[4] Gao, Peng, Shijie Geng, Renrui Zhang, Teli Ma, Rongyao Fang, Yongfeng Zhang, Hongsheng Li, and Yu Qiao. "Clip-adapter: Better vision-language models with feature adapters." arXiv preprint arXiv:2110.04544 (2021).

[5] Sung, Yi-Lin, Jaemin Cho, and Mohit Bansal. "Lst: Ladder side-tuning for parameter and memory efficient transfer learning." arXiv preprint arXiv:2206.06522 (2022).

### Reproducibility
Source code is not attached.

**Strength And Weaknesses:**

## Strengths
- Intuitive approach to utilize pre-trained modules, while it's already known technique in different fields especially NLP.
- Useful observations that show generic pre-trained models might not be sufficient for visual manipulation.
- Performance gain with only introducing a small number of parameters and fine-tuning them instead of whole fine-tuning.

## Weaknesses
- As authors already stated in the related work section, introducing adapter for transfer learning is already known technique and it's not particularly surprising that it also works for robotics tasks (but I think explicitly showing this is still a valuable contribution to the community). It would be more interesting to think of what unique and interesting observations or practical benefits can be achieved by introducing such adapters.
- Following the previous point, more investigation into why we need a loseless adaptation is missing, e.g., what would be the benefit compared to introducing two separate pretrained models and keeping the one while fine-tuning the other one? Or is there an interesting observation that can be made from using adapters for other perception tasks?
- A lot of emphasis is on parameter-efficiency but another important axis would be compute-efficiency. Considered models already seem very large; so it's not clear to me what would be the benefit of being parameter-efficient upon such large models. More discussion on this would be helpful for readers.
- Missing baseline is the performance of training everything from scratch with sufficient compute budgets and varying architecture sizes (as we would not require such a large model for considered downstream tasks).


**Summary Of The Paper:**

This paper introduces an approach that takes pretrained vision modules and fine-tunes them for solving downstream tasks, but importantly without fine-tuning whole parameters by introducing adapters consisting of a small number of parameters. The proposed adapters are designed to be applicable to widely used architectures (ResNet, ViT, NFNet) and the effectiveness of the proposed approach is evaluated on  a wide set of simulator tasks and also sim2real tasks.

**Summary Of The Review:**

While the technical novelty of the proposed approach is not high, it is interesting to see that introducing adapters can be also effective for robotics tasks and claims made in the paper are well supported with exhaustive experimental results. Hence I recommend the paper to be accepted.

---

> ### Author Response · Authors · 2022-11-11
> **Response to Reviewer a8EG**
>
> We thank the reviewer for their detailed comments. We are happy to see they found our paper easy to read and our observations to be useful for the community. Below we address some of the other concerns raised.
>
> **Re:  … what unique and interesting observations or practical benefits can be achieved by … adapters**
>
> Thank you for the suggestion. We have updated our draft with additional observations and practical benefits of using adapters with large pre-trained models. Below we briefly highlight two of them.
>
> **One common model:** Recent works on out-of-domain representation learning for robotics has (Parisi et al. 2022, Nair et al. 2022, Radosavovic et al. 2022) often propose specialized off-the-shelf representations for robot manipulation tasks (Nair et al. 2022,  Radosavovic et al. 2022). Further, they show that these fixed representations can outperform other non-specialized representations such as CLIP/ALIGN.  While we do find similar results for fixed representations, we additionally show that without changing pretrained CLIP/ALIGN weights its representations can be adapted to match full finetuning performance of specialized models (such as MAE used in Radosavovic et al. 2022) and significantly outperform fixed specialized representations. Being able to adapt non-specialized models will allow methods to easily use future large models for manipulation tasks.
>
> **Sim2Real:** To transfer simulation trained policies to real world prior works often use extensive visual domain randomization, photo-realistic simulators, or sensory modalities with little sim2real gap (such as depth). By contrast, our results show that using adapters with large pre-trained models (which are closer to natural image statistics) can avoid the need for any expensive sim2real strategy. We believe this is an important result and can lead to interesting future research especially with larger models trained on more diverse data.
>
>
> **Re: … introducing adapter for transfer learning is already known technique and it's not particularly surprising that it also works for robotics tasks.**
>
> We agree that adapters for transfer learning have been explored in other domains, particularly in the language community. However, since the control setting is substantially different from NLP and vision settings, it is not clear that adapters will work unless it is demonstrated. For example linear probing (a linear trainable head over a frozen model) works great for transfer learning in computer vision settings but the same method is quite suboptimal when used in control settings as we demonstrate in our submission. We think our results provide significant insights for using adapters in control settings. Please also see our elaboration on this point in the general reply to all reviewers.
>
> **Re: emphasis on parameter-efficiency but another important axis would be compute-efficiency.**
>
> Thank you for this interesting suggestion. In the current work we pursue parameter efficiency since it is closest to existing works in control settings (which often only add top adapters with as few parameters as possible). While we agree that compute efficiency is important, especially for real world control tasks, we believe it would require its own separate investigation.
>
> **Re: why we need a lossless adaptation is missing, e.g., what would be the benefit compared to introducing two separate pretrained models and keeping the one while fine-tuning the other one.**
>
> Please see our elaboration on this point in the general reply to all reviewers. We have also updated the exposition in the paper around the need for lossless adaptation for robot manipulation. In short, real world robots will have to solve hundreds or thousands of different manipulation tasks, hence keeping a separate large pre-trained model for each task becomes unscalable. Further, real world robots have limited hardware capabilities which further reduce the number of models that we can keep on the device. Practically, as we move to even larger models (e.g. billions of parameters), lossless adaptation will become more important for resource constrained settings (e.g. robots, edge devices etc.).
>
>
> [1] Parisi et al. The unsurprising effectiveness of pre-trained vision models for control, 2022
> [2] Nair et al. R3m: A universal visual representation for robot manipulation, 2022
> [3] Radosavovic et al. Real-World Robot Learning with Masked Visual Pre-training, 2022

---

> > ### Comment · Reviewer_a8EG · 2022-11-22
> > **Response**
> >
> > Thank you for your detailed response. I don't have a big concern on the paper but still leaving some comments that are not fully resolved by the response, it would be interesting to discuss these stuffs in the paper.
> >
> > - I'm curious to understand how the R3M or MVP representations can be seen as specialized for robotic manipulation, as the only difference might be that they are standard visual models trained on egocentric videos?
> >
> > - It's a bit awkward to state that keeping only one model is important due to resource constraints but it is okay to fine-tune very large models (which leads to slower inference) and not consider compute-efficiency. Should this be the main reason for loseless adaptation, why not we just focus on a direction to train a small but powerful model, or conduct distillation to keep the model small?

---

> > > ### Author Response · Authors · 2022-11-23
> > > **Reply to Response**
> > >
> > > Thank you for your feedback. Below we address both of your questions. We will add the related discussion (as mentioned below) to the paper.
> > >
> > > ***Re: How R3M or MVP representations are specialized for robotic manipulation?***
> > >
> > > Thank you for bringing this to our attention. We referred to R3M (Nair et al.) and MVP (Xiao et al., Radosavovic et al.) representations as specialized for manipulation since they have only been evaluated (and shown to perform well) on manipulation tasks.
> > >
> > > **Dataset choices:** As you pointed out one differentiating factor between these representations and other image representations is the dataset choice. Standard image representations often use ImageNet or web-scale data (CLIP). However, R3M (Nair et al.) uses egocentric videos from Ego4D, MVP (Xiao et al.) uses a curated Human-Object Interaction (HOI) dataset and (Radosavovic et al.) combines ImageNet, HOI and Ego4D. Since human-object interactions from an egocentric perspective mostly focus on human hands and the object being manipulated, it is unclear how generalized the learned visual representations will be.
> > >
> > > **Training choices:** In addition to dataset choices, another differentiating factor between some of these representations (R3M) and general image representations are the training choices. For instance, R3M uses a temporal contrastive objective which is mostly used for control tasks (Sermanet et al. 2018, Dwibedi et al., 2018). Further, as noted in Nair et al, some choices in R3M training were made specifically for control tasks, e.g., R3M directly uses L1 and L2 loss on the representation because they *“.. hypothesize that sparse and compact representations benefit control”* (from Nair et al.). Such choices are not commonplace when learning visual representations from ImageNet (either supervised or self-supervised) or large scale web data, hence their effectiveness for non-control tasks is unclear.
> > >
> > > ***Re: why not we just focus on a direction to train a small but powerful model, or conduct distillation to keep the model small?***
> > >
> > > We definitely agree that compute efficiency – limited compute on real world robots, is an extremely important problem for robot manipulation. We will explicitly add a discussion section to the main paper on the limitations of large vision models (including our approach) in terms of compute efficiency. We will also add a larger discussion on compute efficiency (based on the points below) to Appendix C.
> > >
> > > While it is indeed possible to pre-train and use small models, it is unclear if small models will be as powerful as large models for common sense semantic perception of the world. For instance, recent works have shown that certain model capabilities often only appear for large models (Kaplan et al., Brown et al.). Our aim in this work is to show that robots can utilize models with strong multimodal perception/interaction capabilities (strong semantic understanding, commonsense visual reasoning, open-ended human robot dialog) – while simultaneously adapting the representations from these models for impressive performance on downstream manipulation tasks. While we sacrifice compute efficiency for these abilities, we believe that robots acting in the real world will require such capabilities especially when performing tasks around or in collaboration with humans. That is why we think lossless adaptation of these models is quite important, even if it comes with a cost on compute efficiency. Finally, we do agree it would be important future work to see if small models (pretrained or distilled from large models) can retain the original capabilities of large multimodal models.
> > >
> > > Additionally, we believe our lossless adaptation approach is still important for small models since we try to use as few parameters as possible in our adapter modules. It is possible that further fewer adapter parameters are required for smaller models. Given that we add very few additional parameters our approach barely increases the compute budget compared to the pretrained model (both large and small). While we do not directly investigate compute efficiency we believe our approach combined with the above ideas does provide useful avenues for future research.
> > >
> > >
> > > [1] Nair et al. R3M: A universal visual representation for robot manipulation, 2022
> > > [2] Radosavovic et al. Real-World Robot Learning with Masked Visual Pre-training, 2022
> > > [3] Xiao et al. Masked Visual Pre-training for Motor Control, 2022
> > > [4] Kaplan et al. Scaling Laws for Neural Language Models, 2020
> > > [5] Brown et al. Language models are few-shot learners, 2020
> > > [6] Dwibedi et al. Learning actionable representations from visual observations, 2018

---

### Official Review · Reviewer_eGE9 · 2022-10-24

**Confidence:** 4
**Correctness:** 3
**Technical Novelty And Significance:** 2
**Empirical Novelty And Significance:** 2
**Recommendation:** 6

**Clarity, Quality, Novelty And Reproducibility:**

Clarity

The writing is reasonably clear.

Quality

The execution of the work appears sound.

Novelty

The novelty of the work is low. The pre-training architectures, pre-training objectives, transfer methods of fine-tuning and adapters, and robotic imitation learning benchmarks and datasets are all from prior work.

Reproducibility

The authors provide a reasonable level of detail that would facilitate a solid attempt at reproducibility. Code is not provided or promised, however.

**Strength And Weaknesses:**

Strengths

The authors study the important problem of using pre-trained visual representations for robotics. The empirical evaluation is rather extensive, involving: 5 tasks x 2 domains x 3 camera angles = 30 task instances as used by Nair et al. (2022), as well as 5 task instances from RGB-Stacking; 3 popular architectures for visual representation learning; supervised and self-supervised pre-training objectives; and ablations on the use of adapters in various parts of the pre-trained model. The proposed use of adapters appears to result in performant transfer.

Weaknesses

The motivation of the problem that this paper proposes and tackles is not clear. A simpler and more performant way to solve the lossless adaptation problem would be to do full fine-tuning for adaptation while retaining a separate copy of the pre-trained weights. While I do not have in-depth familiarity with the adapter literature, from a cursory skim it does not seem that these works motivate their proposed methods with losslessness, but rather computational efficiency and memory usage during fine-tuning.

**Summary Of The Paper:**

This work investigates the use of pre-trained visual representations for robotic imitation learning. First, the authors show empirically that transfer should in general involve fine-tuning, as this is shown to dramatically increase performance over using frozen pre-trained weights. Then, to avoid catastrophic forgetting of the pre-training task, the authors study the performance when fine-tuning using "adapters" -- additional layers with small parameter counts that preserve the original network when initialized -- instead of fine-tuning the original weights. The authors show that using adapters can close most of the gap between pre-trained frozen weights and full fine-tuning. They also show that using adapters at multiple points in the model is important for performance.

**Summary Of The Review:**

From this work, I learned that:
1. Using frozen pre-trained visual representations is a relatively ineffective way of doing transfer to robotic imitation learning.
2. Fine-tuning these representations results in very performant i.i.d. transfer.
3. Using adapters gets us close to full fine-tuning.

Overall, this work applies methods from prior work to obtain results that corroborate those found for other domains (Houlsby et al., 2019; [B]). While the execution of the work is good, the limited novelty and unclear significance (specifically regarding "losslessness") makes me lean towards rejection.

References

[A] Kumar et al., Fine-Tuning can Distort Pretrained Features and Underperform Out-of-Distribution, ICLR 2022.

[B] Berriel et al., Budget-Aware Adapters for Multi-Domain Learning, ICCV 2019.

---

> ### Author Response · Authors · 2022-11-11
> **Reply to Reviewer eGE9**
>
> We thank the reviewer for their detailed comments. We are happy to see that you liked our execution. Please find our replies to your comments below:
>
> **Re: The motivation of the problem that this paper proposes and tackles is not clear.**
>
> As elaborated in our general comment to all the reviewers, while fine-tuning the large pretrained model is indeed one way to achieve good performance on downstream manipulation tasks, such an approach is not very scalable for robotic manipulation where the robot will have to solve hundreds of different tasks. This coupled with limited hardware capabilities of real world robots prohibits keeping a separate large model for each task. We hope this alleviates your concern regarding the problem statement. We have also improved our exposition to make the motivation clearer.
>
> **Re: this work applies methods from prior work to obtain results that corroborate those found for other domains.**
>
> Although adapters have previously been used for NLP and to a small extent in vision tasks, their usage in control tasks raises a different set of challenges. We elaborate on these challenges in our general comment to all reviewers. We note that these challenges also necessitate the use of adapters at different locations (e.g. bottom adapters) within the network, as well as sparse adapters in middle sections. By comparison, most prior works (Houlsby et al. 2019, Rebuffi et al. 2017, Li et al. 2022) often only focus on middle adapters. However, as we show through extensive experiments only using middle adapters may not be sufficient to reach close to full fine-tuning performance.
>
> **Re: Unclear significance**
>
> Our work is the first to establish the usefulness of adapting off-domain representations for control tasks in a parameter efficient way. We hope our successful implementation of adapters will allow further exchange of ideas between fields, especially in the context of fine-tuning large pre-trained models. This will be increasingly important for control tasks as large pre-trained models become commonplace.
>
> Finally, we briefly mention some other significant learnings from our results:
> 1. SOTA results using out-of-domain representations for robot manipulation.
> 2. **Sim2Real:** First work to show sim2real transfer without any visual domain randomization on complex manipulation tasks.
> 3. **Specialized vs Non-Specialized Representations:** Show that non-specialized models (CLIP/ALIGN) previously shown to perform poorly can be adapted to outperform representations specialized for manipulation tasks.
>
> **SOTA Results:** Previous works within the learning for control community only adapt pre-trained visual representations using top-adapters. Our work shows that using top-adapters (0.6M params) is insufficient to reach close to optimal task performance. Instead, adding bottom and middle adapters (0.4M params, fewer params than top adapters) significantly improves manipulation performance.
>
> **Sim2Real:** Prior work in robotics often utilize extensive visual domain randomization, photo-realistic simulators, and other sensory modalities with smaller sim2real gap (e.g. depth) to transfer simulation trained policies to real-world. Our results show that using large pre-trained models (which are closer to natural image statistics) can avoid the need for such expensive sim2real strategy. Although compared to visual domain randomization there still exists a performance gap, we believe this opens up significant avenues for future research in robotics.
>
> **Specialized vs Non-Specialized Representations:** Recent works (Nair et al. 2022, Radosavovic et al. 2022) propose specialized representations (specialized for robotics) that can outperform off-the-shelf CLIP representations. Our work shows that while off-the-shelf CLIP representations can be poor, adapting them through our proposed adapters results in similar performance as other adapted representations (such as MAE ones) and *significantly outperforms* fixed off-the-shelf specialized representations. This is important since CLIP/ALIGN representations are more robust (Wortsman et al. 2021) and have been used for dialogue generation, robot navigation. Our result shows that the same pretrained model can perform robot manipulation and achieve great performance. This provides exciting avenues for future research wherein a single fixed model can be used to solve a very wide range of tasks.
>
> We have updated our draft to better reflect the significance of our points.
>
> [1] Rebuffi et al. Learning multiple visual domains with residual adapters, 2017
> [2] Houlsby et al. Parameter-efficient transfer learning for nlp, 2019
> [3] Li et al. Cross-domain Few-shot Learning with Task-specific Adapters, 2022
> [4] Nair et al. R3m: A universal visual representation for robot manipulation, 2022
> [5] Radosavovic et al. Real-World Robot Learning with Masked Visual Pre-training, 2022
> [7] Wortsman et al. Robust fine-tuning of zero-shot models, 2022

---

> > ### Author Response · Authors · 2022-11-17
> > **Can we help address any other concerns?**
> >
> > Dear Reviewer,
> >
> > Thank you for your thoughtful review. Since the paper update phase ends soon, we would like to check if our response has addressed your concerns. Please let us know if there are some other questions we can resolve.
> >
> > Thanks,
> > Paper Authors

---

> > > ### Comment · Reviewer_eGE9 · 2022-11-18
> > > **Review update**
> > >
> > > I appreciate the responses to my suggestions and criticisms. I have a better understanding of the empirical contributions and the point of "losslessness". On the latter, the benefits of efficiency and modularity are rather forward-looking and not concretely studied in the present work, so it might be better-suited for discussion/conclusion rather than in the title/abstract/intro. But this is ultimately up to the authors.
> > >
> > > I have raised my score to a 6.

---

> > > > ### Author Response · Authors · 2022-11-22
> > > > **Thank You**
> > > >
> > > > Thank you for the feedback and increasing your score! We appreciate your suggestion and will consider it in updating the paper.

---

### Official Review · Reviewer_TzMT · 2022-10-24

**Confidence:** 4
**Correctness:** 4
**Technical Novelty And Significance:** 3
**Empirical Novelty And Significance:** 3
**Recommendation:** 8

**Clarity, Quality, Novelty And Reproducibility:**

The paper is clear to read and well written. To the best of my knowledge, proposing and studying this type of lightweight adaptation of pretrained vision models across multiple model layers and architectures is novel. Code doesn't seem to be provided so reproducibility is unclear.

**Strength And Weaknesses:**

Strengths
- Paper is tackling an important and timely challenge of how to best integrate powerful pretrained vision models for control tasks.
- The experimental setup seems to be thorough and well thought out — decomposing the positioning of the adapter modules into top, middle, bottom portions of the network is interesting, and the authors also conduct a thorough comparison across different vision architectures.

Weaknesses
- Would be helpful to get a sense of statistical significance by adding error values to the the success rates reported in Tables 1,2,3 and Figure 5.
- The claim ‘lossless adaptation’ made me think that the authors would investigate how much the model’s representations deviated with the adaptors (i.e. looking at performance changes with respect to original classification, detection capabilities, etc.). Perhaps ‘lossless’ is too strong of a term if not actually demonstrated?
- Paper would be strengthened with including further ablations of their architectural design — e.g. impact of not having the skip connections.

Questions
- Beyond the sim2real experiments, have the authors considered investigating more cross-domain transfer — e.g. how well do the representations adapted for MetaWorld transfer to the Franka kitchen domain? Do the trained adapters transfer well or need to be fully retrained?

**Summary Of The Paper:**

The authors propose a method for finetuning pretrained vision models for robotic manipulation tasks that aims to reduce disruption to the original pretrained visual representation and loss of original representation expressivity. The proposed method injects learned lightweight ‘adapters’ throughout the pretrained architecture at multiple levels, rather than just at the top level. The empirical evaluations show that the approach is able to generally match the success rate of the fully fine-tuned vision representations, but with a fraction of the learnable parameters.

**Summary Of The Review:**

I recommend an 8. The underlying method is straightforward and simple, and the impacts of various types of adaptation, finetuning, etc. seem thoroughly explored experimentally across a range of different environments and a real world task.

---

> ### Author Response · Authors · 2022-11-11
> **Response to Reviewer TzMT**
>
> We thank the reviewer for their comments and encouraging feedback! We are happy to see they found our work to be important and timely and our experiments to be interesting.
>
> **Re: lossless adaptation is not demonstrated**
>
> Thank you for the suggestion. While we do not explicitly run experiments showing the performance of the large pre-trained model before and after adaptation, we ensure that none of the pre-trained weights are updated while training on the downstream manipulation task. Although this theoretically ensures that the existing capabilities of the model remain intact it does have one limitation, i.e., this does not allow simultaneous inference over the pretraining and the downstream task. Specifically, we need to enable/disable behaviors to get results for the pretraining and manipulation tasks.
>
> **Re: Paper would be strengthened with including further ablations of their architectural design**
>
> We agree that further ablation studies on the architecture design such as the importance of skip connections would be useful. While we do not perform ablation for skip connections on control tasks, we note that prior work on adapters (Rebuffi et al 2017) does explore serial adapters and found them to be less performant for multi-domain image classification.
>
> **Re: Beyond the sim2real experiments, have the authors considered investigating more cross-domain transfer**
>
> Thank you for this interesting suggestion. We did not explicitly consider cross-domain transfer for the environment suites. One challenge for such transfer across our current environments suites are the suite-specific camera locations which result in very different views between environments. See Figure 4 in the paper which shows views from one camera location for each suite. Such different non-canonical views were unlikely to be present in the pre-training data. Moreover, since we use a single task training setting it is possible that the environment-specific representation overfits to the provided view, thus making cross-domain transfer harder. Although we agree that this would be a great investigation it would require its own comprehensive research on viewpoint and embodiment invariance/equivariance.
>
> [1] Rebuffi et al. Learning multiple visual domains with residual adapters, 2017

---

### Official Review · Reviewer_1MbS · 2022-10-24

**Confidence:** 3
**Clarity, Quality, Novelty And Reproducibility:** See above for originality. The work i…
**Correctness:** 3
**Technical Novelty And Significance:** 2
**Empirical Novelty And Significance:** 2
**Recommendation:** 5

**Strength And Weaknesses:**

## Strengths
- Evaluation of the method itself is comprehensive and the results are meaningful.
- The sim2real results are impressive.

## Weaknesses
- The following is untrue: "While prior work on robotic manipulation has predominantly used frozen pretrained features, we demonstrate that in robotics, unlike in other domains, this approach can fail to reach optimal performance, and that fine-tuning of the full model can lead to significantly better results." In other domains, fine-tuning also improves performance. For example, in ImageNet classification, frozen pre-trained features performance worse than fine-tuning the full model. This is shown by the difference in fine-tuning and the linear probe evaluation in recent SSL papers, for example "Masked Autoencoders Are Scalable Vision Learners, He et al, CVPR 2022".
- While the authors do a good job demonstrating that adapters are effective at closing the gap between fine-tuning and frozen, it is unclear to the reviewer if the proposed adapters are novel. The proposed adapters are very similar to those in "Cross-domain Few-shot Learning with Task-specific Adapters, Li et al, CVPR 2022". It appears as they the exact formulation of the adapters is different however no comparison in performance is provided.

**Summary Of The Paper:**

This paper studies the problem of how to leverage pre-trained visual representations for control. They show that simply freezing the visual encoder and training a policy head leads to poor performance. Performance can be improved by fine-tuning the visual encoder but this leads to catestrophic forgetting and thus a seperate set of (large) weights must be stored for each task.

The authors propose to use adapters that add a small number of additional parameters to the model and allow it to be adapted to a new task. Only the paremeters of the adapters are learned during fine-tuning, thus the same set of visual encoder weights can be acrossed mutliple tasks.

The proposed method is evaluated across 3 sets of tasks, Metaworld, Kitchen, and RGB block stacking using 3 different backbones. The proposed method is able to (mostly) close the gap to full fine-tuning in all cases. The proposed adapters are also effective in zero-shot sim2real.

**Summary Of The Review:**

My primary concern with this paper is the evaluation of the proposed adapters. While it is clear that they are effective, it is unclear how they compare with similar methods already present in the literature.

---

> ### Author Response · Authors · 2022-11-11
> **Reply to Reviewer 1MbS**
>
> We thank the reviewer for their comments. We are encouraged to see that they found our results to be meaningful and our sim2real results to be impressive. Please find our replies to your comments below:
>
> **Re: it is unclear to the reviewer if the proposed adapters are novel**
>
> As we note in the general comment to reviewers above, our usage of adapters is similar to previous works (Rebuffi et al., Houlsby et al.). However, unlike previous works we use adapters for the control problem which faces a different set of challenges and thus, necessitate its own investigation. Please see our elaboration on this point in the general reply to all reviewers. Briefly, significant *task-differences* (between pre-training and downstream tasks) coupled with *domain shifts* require carefully inserting adapters in different sections without tremendously increasing the parameter count. For instance, we show bottom adapters are important and sparse adapters in middle sections are sufficient.  We have also updated the paper to better reflect these differences.
>
> Overall, our contribution is primarily to implement adapters in a manner appropriate for the robotic manipulation setting, and to carefully evaluate their use in this setting. We provide a comprehensive evaluation and demonstrate that adapters can significantly outperform the use of frozen representations which is common in the community.  Moreover, specific to robotic settings, we demonstrate that adapters have significant sim2real transfer results.
>
>
> **Re: Similarity with adapters used in "Cross-domain Few-shot Learning with Task-specific Adapters, Li et al, CVPR 2022**
>
> Both Li et al. 2022 and our work considers similar convolutional adapters which are also similar to the original adapters proposed in Rebuffi et al. 2017. The main difference is that Li et al. 2022 solve few-shot image classification whereas we solve robot control problems which brings many other challenges to be addressed as described in our general response. Additionally we also use adapters in bottom/top sections, while Li et al. 2022 only use adapters in middle sections. In that sense our middle adapter only results can be seen as a comparison to their method but it wouldn’t be a fair comparison as they are not solving a control problem.
>
> **Re: Comparison to methods already present in the literature**
>
> Since our work introduces the idea of adapters to the control community, we compare against existing methods in the control literature. However, approaches in the control literature only use fixed off-the-shelf representations (or use full fine-tuning) and we do compare against them in our work. Among the adapter methods in the language and vision community, we use the initial formulation of adapters (Houlsby et al. 2019 for transformers, Rebuffi et al. 2017 for convolutional networks), which we found to work well in our robotic manipulation setting.
>
> **Re: The following is untrue: “While prior work …, unlike in other domains, …**
>
> Thank you for pointing this out, and we indeed agree that even in other domains/tasks such as ImageNet classification, full finetuning on pre-trained features can result in better performance than linear probing. We have updated the text to reflect this.
>
>
> [1] Rebuffi et al. Learning multiple visual domains with residual adapters, 2017
> [2] Houlsby et al. Parameter-efficient transfer learning for nlp, 2019
> [3] Li et al. Cross-domain Few-shot Learning with Task-specific Adapters, 2022

---

> > ### Comment · Reviewer_1MbS · 2022-11-16
> > **Reply**
> >
> > > Since our work introduces the idea of adapters to the control community, we compare against existing methods in the control literature.
> >
> > This is great and necessary. However, given the rich literature of adapters that already exists, even though it's all in different domains, explaining to the reader how your adapters differ and demonstrating that these differences matter is also necessary. Without this the reader is left wondering if any adapter method would achieve the same results or if there is something specific to control in your method.
> >
> > > Additionally we also use adapters in bottom/top sections, while Li et al. 2022 only use adapters in middle sections. In that sense our middle adapter only results can be seen as a comparison to their method but it wouldn’t be a fair comparison as they are not solving a control problem.
> >
> > This is what I was looking for and I suggest the authors make sure this comes across in paper. I agree that Li et al. 2022 designed their method for a different task, but the fact that their method isn't suitable for control problems, while yours is, is a key piece of information.
> >
> > If I am understanding things correctly, the proposed adapters are novel because they are a combination of top, bottom, and middle adapters?

---

> > > ### Author Response · Authors · 2022-11-16
> > > **Re: Similarity with adapters used in "Cross-domain Few-shot Learning with Task-specific Adapters, Li et al, CVPR 2022**
> > >
> > > Thank you for your feedback. Yes your understanding is correct, prior works (Rebuffi et al 2017, Li et al. 2022) do not consider bottom or top adapters (for their classification task), however as we show in Section-5.2 using adapters in these sections is important for the control tasks. We have also updated our draft (e.g. Page: 3 Para: 2) to explicitly mention the differences between our work and related works. Based on your above suggestion, we have also updated (Page 5, Para: 2 and Page 8, Para: 2 (bottom) and Page 17, last para) to reiterate this difference.
> > >
> > > Please let us know if you have any further questions or clarifications.

---

### Author Response · Authors · 2022-11-11
**General Reply to All Reviewers**

We thank the reviewers for their feedback and detailed comments. We are encouraged to see that two reviewers advocate for acceptance, noting the importance of the problem we explore and the thoroughness of our results (Reviewer TzMT), and showing that adapters work on robotic tasks is a valuable contribution to the community (Reviewer a8EG).

While Reviewers 1MbS and eGE9 assigned borderline low scores, we believe their concerns can be resolved in the rebuttal period and addressed through improvements in our exposition. To sum up:
- Reviewer 1MbS is unclear how our proposed adapters compare with similar methods already present in the literature.
- Reviewer eGE9 is unclear about the significance of our problem/approach specifically around lossless adaptation for robot manipulation.

Before addressing each reviewer’s individual concerns in their respective threads, we briefly address some common points below (we have also updated the paper accordingly):

***Why lossless adaptation?:*** We propose lossless adaptation of pretrained vision models based on crucial practical considerations in robot manipulation. As general purpose vision models (e.g. CLIP/ALIGN) become more powerful; they will be used increasingly in robotic settings e.g. human-robot interaction  (human-robot dialog with visual context), describing and *solving* robotic tasks. However these models are prohibitively large to have multiple task-specific copies on many resource limited settings (e.g. biped, quadruped robots, mobile manipulators etc.). More so, as evidenced by recent results these models will become much bigger over time (new VLMs have billions of parameters). Thus, fine-tuning the original large pre-trained model for each new robotic task and keeping another copy (as suggested by Reviewer eGE9) to preserve existing vision-language capabilities is highly resource intensive. It is especially unscalable for general purpose robots which will end up solving thousands of downstream manipulation tasks (e.g. even current benchmarks such as MetaWorld, which are very limited when compared to human manipulation abilities, already have 50 different tasks). That is why lossless adaptation is a very important problem in the context of robotics (and other resource limited settings, e.g. smart devices, etc.).


***Adapters for control tasks:*** A major contribution of our paper is to introduce adapter modules for control tasks. Adapters have been explored within the language community and to a far lesser extent in the vision community. However, since the control tasks represent a different set of challenges (discussed below), the effectiveness of adapters is unclear unless demonstrated. We also hope our contribution will allow for further cross-pollination of ideas between these communities. Below, we briefly discuss these differences:


- **Task Differences:**  Prior works often utilize adapter modules for a broadly similar set of tasks. For instance, text classification (Houlsby et al., 2019), neural machine translation (Pham et al., 2020), or few-shot visual classification (Li et al., 2022). Additionally, the pretrained model used in these tasks (e.g. ImageNet pretraining) is strongly correlated with the downstream tasks (few shot classification). However, in our work, the visual pretraining task (e.g. image-inpainting/classification) is vastly different from robot manipulation (continuous action prediction). While visual classification tasks mostly require *semantic* features (i.e. “what” is in an image), manipulation tasks require *actionable* features (“what” and “where” are the objects to predict actions) (Dwibedi et al., 2018; Rosinol et al., 2020).  These actionable features need to carry both semantic as well as *spatial* information (e.g. being able to localize objects of interest in space).


- **Domain Shifts:** Few works in the language community use adapters for domain shifts (domain adaptation) (Zhang et al.,2021) (even (Zhang et al., 2021) use unlabelled target data to pretrain adapters).  Works in the vision community do involve domain shifts, e.g., training on ImageNet, Birds and transfer to MS-COCO, CIFAR-10 (Li et al., 2022).  However, these shifts are very different from those encountered in our work: robot manipulation data consists of table-top settings, simulation images, moving robot arm, heavy occlusion, textureless objects which is very different from object-centered images of ImageNet often used for pretraining.


[1] Rebuffi et al. Learning multiple visual domains with residual adapters, 2017
[2] Houlsby et al. Parameter-efficient transfer learning for nlp, 2019
[3] Li et al. Cross-domain Few-shot Learning with Task-specific Adapters, 2022
[4] Zhang et al. Unsupervised Domain Adaptation with Adapter, 2021
[5] Dwibedi et al. Learning actionable representations from visual observations, 2018
[6] Rosinol et al. 3D dynamic scene graphs: Actionable spatial perception with places, objects, and humans, 2020

---

### Decision · Program_Chairs · 2023-01-20

**Decision:**

Accept: poster

**Justification For Why Not Higher Score:**

 Only small algorithmic novelty.

**Justification For Why Not Lower Score:**

Impressive evaluations.

**Metareview: Summary, Strengths And Weaknesses:**

The paper presents an extensive study of adaptation mechanisms for representation learning in the context of robot manipulation.

The paper offers exhaustive experiments to evaluate the effectiveness of different adaptation schemes for representation learning in the context of robot manipulation. The experiments are impressive and well executed and show interesting results. The paper is addressing a timely challenge of how to best integrate powerful pretrained vision models for control tasks. There were some initial issues regarding novelty of the approach and distinction to related methods. However, this was addressed adequately by the rebuttal of the authors as the paper is considering a different problem domain which also requires different adaptation architectures. I consider most concerns of the reviewers to be resolved adequately and recommend acceptance.

**Note From Pc:**

if the above contains the word "oral" or "spotlight" please see: "oral" presentation means -> notable-top-5% and "spotlight" means -> notable-top-25%. As stated in our emails, we are disassociating presentation type from AC recommendations

**Summary Of Ac-Reviewer Meeting:**

N/A